# Strategies adopted by men to deal with uncertainty and anxiety when following an active surveillance/monitoring protocol for localised prostate cancer and implications for care: a longitudinal qualitative study embedded within the ProtecT trial

Julia Wade [1], Jenny Donovan,[1] Athene Lane,[1] Michael Davis,[1] Eleanor Walsh,[1] David Neal,[2] Emma Turner,[1] Richard Martin,[1] Chris Metcalfe,[1] Tim Peters,[1] Freddie Hamdy,[2] Roger Kockelbergh,[3] James Catto,[4] Alan Paul,[5] Peter Holding,[2] Derek Rosario,[6] Howard Kynaston,[7] Edward Rowe,[8] Owen Hughes,[9] Prasad Bollina,[10] David Gillatt,[11] Alan Doherty,[12] Vincent J Gnanapragasam,[13,14] Edgar Paez[15]

For numbered affiliations see end of article.

**Correspondence to**
Dr Julia Wade;
julia.wade@bristol.ac.uk

## ABSTRACT

**Objectives** Active surveillance (AS) enables men with low risk, localised prostate cancer (PCa) to avoid radical treatment unless progression occurs; lack of reliable AS protocols to determine progression leaves uncertainties for men and clinicians. This study investigated men's strategies for coping with the uncertainties of active monitoring (AM, a surveillance strategy within the Prostate testing for cancer and Treatment, ProtecT trial) over the longer term and implications for optimising supportive care.

**Design** Longitudinal serial in-depth qualitative interviews every 2–3 years for a median 7 (range 6–14) years following diagnosis.

**Setting** Four centres within the UK Protect trial.

**Participants** Purposive sample of 20 men with localised PCa: median age at diagnosis 64 years (range 52–68); 15 (75%) had low-risk PCa; 12 randomly allocated to, 8 choosing AM. Eleven men continued with AM throughout the study period (median 7 years). Nine received radical treatment after a median 4 years (range 0.8–13.8 years).

**Intervention** AM: 3-monthly serum prostate-specific antigen (PSA)-level assessment (year 1), 6–12 monthly thereafter; increase in PSA ≥50% during previous 12 months or patient/clinician concern triggered review.

**Main outcomes** Thematic analysis of 73 interviews identified strategies to accommodate uncertainty and anxiety of living with untreated cancer; implications for patient care.

**Results** Men sought clarity, control or reassurance, with contextual factors mediating individual responses. Trust in the clinical team was critical for men in balancing anxiety and facilitating successful management change/continued

## Strengths and limitations of this study

► This is the first longitudinal qualitative study to document men's longer-term experiences of an active surveillance protocol for localised prostate cancer, with 73 interviews conducted over a period of 6–14 years.

► Although the sample size was small (20 men), sampling was led by principles of maximum variation in line with guidelines for qualitative research and enabled identification of commonalities and differences across groups.

► Findings provide insights into how clinical care for men following active surveillance protocols can be tailored to support individuals.

► Men following active surveillance protocols outside the Prostate testing for cancer and Treatment trial may have different experiences.

► Few men were of a non-white background.

monitoring. Only men from ProtecT were included; men outside ProtecT may have different experiences.

**Conclusion** Men looked to clinicians for clarity, control and reassurance. Where provided, men felt comfortable continuing AM or having radical treatments when indicated. Clinicians build patient trust by clearly describing uncertainties, allowing patients control wherever possible and being aware of how context influences individual responses. Insights indicate need for supportive services to build trust and patient engagement over the long term.

**Trial registration number** ISRCTN20141297; Pre-results.

## BACKGROUND

Uptake of active surveillance/monitoring (AS/AM) as a management strategy for low-risk prostate cancer (PCa) has increased markedly in recent years.[1–3] AS/AM offers the opportunity to delay or avoid significant side effects associated with radical treatments.[4–6] Yet protocols for AS/AM vary and the optimum approach to conservative management remains contested, with programmes variably including serum prostate-specific antigen (PSA) testing, digital rectal examination and/or prostate biopsy with or without multiparametric MRI (mpMRI), with the aim of detecting clinical progression for timely curative radical treatment where indicated.[7–9]

AS/AM protocols require men to live with uncertainty about the consequences of leaving the cancer untreated. Current clinical protocols adopt two premises: first, men can request radical treatment if they no longer tolerate this uncertainty and/or there is evidence of clinical progression; second, this 'window of opportunity' for cure may be missed. Up to half of men following an AS/AM protocol change to radical treatments within 10 years[7–10]; 30% of changes occur without indication of clinical progression.[11–13] Systematic reviews (SRs) of the literature present a mixed picture regarding psychological well-being of men undergoing AS/AM[14–17]; while most men report positive psychological well-being, a minority report short-term heightened anxiety, uncertainty and distress. A recent SR of barriers and facilitators to selecting and adherence to AS[18] reported a range of factors affecting adherence, leaving little clarity on how to optimise the strategy. Cross-sectional studies report men following AS normalise the cancer by 'screening off' the diagnosis[19] or reframing its threat[19 20] or actively engage in lifestyle changes to 'do something extra.'[19 21] Although there is evidence that anxiety is minimal[22] or decreases with time on AS,[23] longitudinal qualitative research on long-term experiences of AS remains scarce. Yet longitudinal studies offer an essential insight into how participants' needs and experiences may change over time, particularly as the disease develops slowly and patients may need to consider remaining on it or changing management over very many years. Longitudinal research also enables exploration of sensitive issues as the relationship between interviewer and interviewee develops.[24]

We addressed this evidence gap by conducting a longitudinal qualitative investigation of the experiences of men who followed the active monitoring (AM) protocol as their primary treatment in the Prostate testing for cancer and Treatment (ProtecT) randomised controlled trial (RCT)[25] over a median of 7 (range 6–14) years. We investigated men's experiences of uncertainties and strategies to manage this. Findings provide insights to shape clinical services for the increasing numbers of patients now undergoing AS/AM programmes for clinically localised PCa.

## METHODS
### ProtecT study

The ProtecT RCT compared radical prostatectomy (RP), radical radiotherapy (RT) and AM for clinically localised PCa diagnosed following PSA testing in men identified in primary care.[6 9 25] Men declining randomisation were offered identical follow-up, forming a comprehensive cohort.[26] ProtecT AM aimed to allow men to avoid or delay radical treatment and its side effects unless their PCa progressed. It involved monitoring with PSA tests every 3 months for 1 year, then every 6 or 12 months thereafter, with an annual review. If PSA rose by 50% or more in a 12-month period or any concerns were raised by the patient or clinician at any time, there could be a re-evaluation of cancer status which could lead to continuing AM or changing to a radical option. This is described in detail elsewhere.[9 25] A model of nurse-led, urologist-supported care was developed for delivery of AM.[27] Men were informed of uncertainties: while the intention was to offer radical treatment when evidence of cancer progression was identified, progression might occur undetected and opportunity for cure be missed; and there was little evidence on effective strategies for monitoring. Uncertainties with AM were expected to be similar to those experienced with other AS strategies because of the lack of international consensus on protocols.[28]

### Longitudinal interview study

This study was part of a larger, longitudinal interview study, investigating men's experiences of treatment received within the ProtecT study and including participants following each of the treatment pathways (AM, RT, RP) with a median follow-up of 10 years.[29] In total 88 men were invited, and 65 men took part in one or more interviews. Experiences of men following an RT pathway have been submitted for publication elsewhere.[30]

### Participants

Of those recruited to the longitudinal interview study, 20 men were recruited at diagnosis, chose or were randomly allocated to follow an AM pathway and took part in three or more interviews; their data are reported here. ProtecT study participants undergoing AM as their primary management following random allocation or treatment choice and within 12 months of PCa diagnosis were eligible for this prospective interview study; this included men with clinically localised or intermediate/high-risk disease in line with eligibility criteria for the ProtecT RCT.[29] Purposive sampling was used with the aim of obtaining a maximum variation sample for this interview study in terms of participant characteristics identified a priori as potentially impacting on experiences: age, socioeconomic background, whether randomly allocated to or choosing AM, and with low-risk or intermediate/high-risk disease. Recruitment took place at four of nine ProtecT centres to capture as diverse a range of experience as possible, while limiting time required for travel by the interviewer. Three centres were located across England and one was in Wales, UK.

Men received a written invitation explaining the purpose of the longitudinal interview study embedded within the RCT, followed by a telephone call establishing

willingness to participate.[29] Five men consenting to the interview study at time of PSA testing were diagnosed with localised PCa and underwent AM. Twenty-one further men were invited following diagnosis and random allocation or treatment choice; 15 agreed, giving a total of 20 participants consenting to take part in the interview study (table 1). Two men refused, four did not respond and six were ineligible due to receiving treatment outside the ProtecT study or because more than 12 months had passed since management by AM had begun.

### Data collection

First interviews were scheduled around 6 months after men started AM, subsequent interviews at 2–3 yearly intervals. Data collection commenced in 2000 and final interviews took place in 2014. Most were conducted by JW, a social scientist with experience in qualitative health

services research, with a small number conducted by LS, a research nurse within the ProtecT study or by LB, a social scientist with experience in qualitative health services research (see the acknowledgements section). All initial interviews took place face to face to put interviewees at ease, in the location of their choice in homes, hospital outpatient departments or university premises as preferred. Subsequent interviews were face to face or by telephone according to interviewee preference. Interviews lasted an average of 50 min (range 8–123 min). All were audio-recorded with consent and transcribed verbatim, at which point identifying data were removed or disguised. Partners could contribute where requested, following the same consent process.

Interview topics are shown in online supplementary appendix A. These were derived from the literature prior to data collection but subsequently informed by data collection. Topic guides were tailored to fit experiences, for example, questions regarding experiences of radical treatments were included for those changing management. Interviews were therefore in-depth and flexible, with prompts used to explore areas of interest, while allowing individuals to introduce content of relevance to them. Field notes were made in tandem with audiodata collection and analysed in parallel with transcript data.

### Data analysis

Thematic analysis was used, applying principles of constant comparison based on grounded theory.[31] Data collection and analysis proceeded iteratively, enabling preliminary findings to inform further data collection. Eight initial interview transcripts were coded and themes identified independently by JW and LS (see acknowledgements) for subsequent cross-comparison of codes/themes, to refine the topic guide and inform further sampling and data collection. Disagreements were resolved by discussion to reach consensus. Analysis proceeded by comparing data between groups within the sample to identify commonalities and contrasts of experience according to group. Groups included those accepting random allocation or choosing AM; remaining on AM or changing to radical treatment; with lower and higher risk disease; older or younger at diagnosis; and in earlier and later interviews to explore how views and experiences evolved over time. Data analysis was supported by qualitative data software NVivo.[32] Coding carried out in NVivo enabled rapid retrieval of coded data for systematic comparison across participants and time.[32 33] The standards for reporting qualitative research[33] were applied in reporting this study. Interpretations of study data were regularly discussed by JW and JD to counter risk of overinterpretation of findings.

### Patient and public involvement

The ProtecT study was awarded funding in 1999, with further funding awarded in 2006, at a time when patient and public involvement (PPI) was relatively new and the original protocol did not explicitly outline PPI

| Table 1 Characteristics of interview study participants n=20 | | |
|---|---|---|
| | Accepted random allocation | Chose treatment |
| | n=12 | n=8 |
| Age | | |
| 50–59 years | 2 | 1 |
| 60–69 years | 10 | 7 |
| Social class | | |
| Managerial/professional | 5 | 4 |
| Other | 7 | 3 |
| Missing | 0 | 1 |
| Study centre | | |
| 1 | 1 | 1 |
| 2 | 3 | 0 |
| 3 | 6 | 7 |
| 4 | 2 | 0 |
| D'Amico risk category* | | |
| Low | 8 | 7 |
| Intermediate/high | 3 | 1 |
| Timing of switch to radical treatment (n=9) | | |
| 0–5 years | 2 | 3 |
| >5 years | 2 | 1 |
| >10 years | 1 | 0 |
| N interviews during AM | | |
| 1 | 0 | 1 |
| 2 | 3 | 1 |
| 3 | 8 | 5 |
| 4 | 0 | 0 |
| 5 | 0 | 1 |
| 6 | 1 | 0 |

*Data missing for one participant.
AM, active monitoring.

work. However, early qualitative work led to changes in patient information provision.[34] In addition to findings reported here, qualitative interviews elicited participant experiences on study processes resulting in changes, for example, making patient questionnaires available electronically. Study participants contributed to written patient summaries of the ProtecT trial findings disseminated in 2016[6 9] and key findings of the study were fed back to ProtecT study participants (including some who took part in the interview study) in face-to-face meetings in June 2017, when participants were consulted on how to optimise further dissemination of the trial findings to men newly diagnosed with PCa.

## RESULTS

Twenty participants (median age at diagnosis 64 range 52–68 years) were interviewed a minimum three and maximum six times, with 73 interviews conducted over a median of 7 (range 6–14) years follow-up. Nine (45%, 5 with low-risk disease) underwent radical treatment after a median of 4 years (range 9mo-13yr9mo) follow-up, comparable with the ProtecT study[9] Eleven (10 with low-risk disease) continued with AM for a median of 7 years (range 6–7 years) follow-up (table 1). Comparison of baseline characteristics to those of men in the main ProtecT study[35] showed broad similarities: median age was 64year (main study: 63year); 9/20 (45%) were in managerial occupations (main study: 46%); 15/20 (75%) had low-risk disease (main study: 72%).

Comparison of experiences across these groups showed more commonalities than differences, suggesting that the themes were relevant to men following AM. Five men with intermediate/high-risk disease expressed similar views to those with low-risk disease. There was a suggestion that higher risk men changed management sooner than those with low-risk disease, but this will be more comprehensively investigated in a future ProtecT study paper.

Men highlighted the paradox of receiving a potentially life-threatening diagnosis without experiencing symptoms. Some viewed this positively: there was no functional impact on their life. For most it brought uncertainty, the possibility of cancer progression without tell-tale signs. Uncertainty about future cancer progression merged with uncertainty about the ability of AM to detect progressing disease in time to initiate curative treatment (table 2). Men responded by looking for clarity, control and reassurance. Contextual factors mediated choice of response for individual men and over time. Where men's needs were met, trust in health professionals grew.

### Strategies: seeking clarity, control and reassurance

Men employed strategies broadly categorised under three themes: clarity, control and reassurance. Themes and strategies were partly complementary and yet potentially in tension: the active search for clarity or control could potentially conflict with the acceptance required to take on reassurances and superficially opposing behaviour might be motivated by the same underlying desire for control (table 3). Strategies were employed iteratively and in parallel, with reliance on specific strategies varying between individuals and over time (figure 1). Seeking clarity and control were strategies often employed early in the monitoring process. Men expressed a greater sense of reassurance as time passed, unless they lost confidence in the monitoring process (see below).

At diagnosis, the most common strategy was to gain clarity to reduce uncertainty. Men questioned health

| Table 2 | Response to diagnosis: managing uncertainty |
|---|---|
| Absence of symptoms | I just carried on a normal life. And my work and my well-being wasn't affected. I had none of the classic symptoms at all.<br>P8-Pref-H, Int +3mo, radical treatment +2yr11mo |
| Uncertainty about prognosis | You've got this thing growing inside you and you know, it's not as if it's somewhere like outside of you where you can feel a lump or something like that, you just don't know what's going off….I think that it would be true to say that I feel my life has been invaded, I view the cancer as like, an invasion of my life.<br>P6-Pref-L, Int +1yr9mo, radical treatment +8yr11mo |
| | When I first went for the active monitoring I thought well, this isn't going to be a problem, I'll be able to sail through this. Because I know that it's contained and I know I'm being monitored, so if there's anything that suddenly changes, which needs a different route, then we're there, we're ahead of the game. But then it wasn't, in reality, as simple as that, because, I then knew, because I'd been diagnosed, that I had cancer and it messes things up a bit, you know? You think you can handle it, but it's always there niggling away in your mind.<br>P19-Pref-L, Int +7mo, continued AM |
| Uncertainty about AM protocol | What's the danger level any rate? What is the danger level? So, I don't really know I mean they say that by this blood test they can tell if it's suddenly come active or not. I don't know, can they?<br>P18-Pref-L, Int +6mo, withdrew from AM+7year |

continued AM, continued AM throughout period of follow-up; H, high risk; I:, Interviewer; I, intermediate risk; Int+1yr1mo, timing of interview was 1-year 1-month postdiagnosis; L, low risk; P:, patient; Pref, chose treatment; Radical treatment+4yr4mo, radical treatment initiated 4years 4 months postdiagnosis; Rx, randomly allocated to treatment; Unknown, missing data; W:, patient's wife; withdrew from AM +7yr post diagnosis, withdrew from AM 7years post diagnosis.

**Table 3** Strategies for managing uncertainty: seeking clarity, control and reassurance

**Clarity**

| | |
|---|---|
| Search for clarity on triggers for active treatment repeatedly foregrounds uncertainty and the need to trust | P: [Research nurse] said 'We've got parameters to follow and if those parameters are fulfilled—or you fall outside those parameters, we would—the alarm bells would ring and we'd do something about it'. I suppose I should be reassured with that.<br>I: Are you clear on what those parameters are? Or, do you feel like you know what those parameters are?<br>P: No, no, no.<br>P11-Rx-L, Int +2yr1mo, continued AM |
| | The basic reason the ProtecT study is running is because there is no kind of definite knowledge about the outcome of either of the treatments and yet that is what it is all about. And that is all that the ProtecT study is trying to deduce. So, everybody can be only as helpful as they can be. They can't then tell me things that they don't know.<br>P20-Pref-L, Int +5mo, continued AM |

**Control**

| | |
|---|---|
| Careful choice of language showing influence of words used by health professionals. | He [urologist] said, 'Believe you me, at the moment it's dormant' and he said, 'Personally speaking he said, I don't think it's anything to worry about'. He said you may have had that a long, long, long, long time<br>P18-Pref-L, Int +6mo, withdrew from AM+7yrWell, it was explained to me it was only a small, small bit you know, nothing to worry about, it hadn't spread, or weren't going to spread or you know, at least that's the sort of words that they used.<br>P16-Rx-L, Int +2yr1mo, continued AM |
| Choosing who to tell/when to talk about diagnosis | We never told anybody, except I think we told one of the children. I thought we ought to do that, I wouldn't tell anybody at all. P11-Rx-L, Int +6mo, continued AM |
| | I don't want all and sundry knowing, you know, that this is around because everybody would be coming up to me and saying, 'How're you doing?' or whatever else. Those that I have told know how I feel about it, I've told them that I don't want to talk about it, that it's just one of those things I've got to get on with and should I deteriorate to the point where it affects them, I'll tell them.<br>P4-Rx-I, Int+9mo, Radical treatment +4yr5mo |
| | I don't talk to people about it, I don't like to. I like to call myself healthy 'cos I think talking about it pulls me down a little bit, with mates and that.<br>P17-Pref-L, Int +2yr1mo, continued AM |
| Determining frequency of PSA tests: –3-monthly tests to act fast if needed. –6-monthly tests to control how often thoughts of PCa intrude. | **Preference for 3 monthly PSA tests**<br><br>I'd rather have it taken every three months just to make sure, where we are. I wouldn't like to go out further than three monthly, at the moment. I have been asked if I wanted it to go out for six monthly but no, the way things are, I'd rather keep it 3 monthly.<br>P12-Rx-L, Int +1yr11mo, continued AM<br><br>**Preference for 6-monthly tests**<br><br>I have to put it away in a box and put it away for every three months and then you think about it every three months, you know? So, six months is good because you've got a longer period of time, now. And it just goes, you don't think about it day to day.<br>P13-Rx-L, Int +2yr1mo, continued AM<br><br>**Timing of test relative to start of month**<br><br>I've been coming in for the test at the end of the month and I've just asked can my appointment be put at the beginning of the next month because psychologically, you know. During May, waiting for the 22nd of May, I've not been too clever. That maybe answers some of how I am feeling, you know, that month is not good and yet I've got to wait now til next week to find the result but, you know, it's very, a very anxious time.<br>P4-Rx-I, Int+9mo, radical treatment +4yr5mo |
| Choosing which healthcare professional is seen or whether face-to-face or telephone. Preference for face-to-face contact Evidence of change with time as trust increased | **Preference for face-to-face visits**<br><br>P: Because they said, they could do it on the phone, but I said, I don't mind going over there.<br><br>I: So is that that you actually prefer to go over there than do it on the phone?<br><br>P: I do prefer to go over to be honest, well I just feel more, more at ease<br>P17-Pref-L, Int +2yr1mo, continued AM<br><br>Contrast comment 2yr10mo later<br>She said I think you could go for 6 months now, so I agreed with her<br>P17-Pref-L, Int +4yr11mo, continued AM<br><br>I deal with the same person all the time. But that's just me. And the phone didn't work<br>P13-Rx-L,+2yr1mo, continued AM<br><br>Contrast with comment 3 year later<br>The nurse said give it [telephone review] a try, she said it's about time, basically, and she said you will give it a try and I'll ring you at that time on that day and she rang me at that time on that day, so it worked quite well actually, because I'm not one to say a lot on the phone<br>P13-Rx-L, Int +5yr1mo, continued AM |

Continued

**Table 3** Continued

| Plotting PSA values on graphs to 'demonstrate a trend' | Once a year I go to [hospital] and they give me a graph there to follow the route of things<br>P14-Rx-L, Int +4yr9mo, continued AM |
| --- | --- |
| | It is demonstrating a trend. You see as a [profession] I do a lot of monitoring of [physical trend]. Now I have the same sort of problems with [physical trend]. They either open or close all the time, which is then not a problem, because it is not getting worse, but if there is a trend then you know that you need intervention to stop it from going off.<br>P19-Pref-L, Int +4yr11mo, continued AM |
| Controlling diet or engaging in exercise or wanting info on what diet changes to make | I won't go for the steak, definitely not, it'll be more, 9 times out of 10 now it'll be the fish, so it's that that kind of change really. It's definitely kicked in since [diagnosis]<br>P4-Rx-I, Int+9mo, radical treatment +4yr5mo |
| | P: I started taking some Lycopene tablets |
| | W: And also buying tins of tomatoes. Just read bits and pieces. And broccoli. |
| | P: I've always ate broccoli |
| | W: I know but I'm just trying to push it in a bit more. But I suppose If you're being honest, we have looked at diet<br>P13-Rx-L, Int +4mo, continued AM |
| **Reassurance** | |
| Looking for reassurance and developing trust | Knowing, I think knowing once you're in the, in the circle, you know you're in with, this system, you feel perhaps that you're part of it and that, you know, everyone says that if you get any problems just ring us up, and you can, you know, you can be done tomorrow<br>P9-Pref-L, Int +5mo, radical treatment +3yr4mo |
| | I know the ultimate decision is mine whichever way we go. But I usually let them lead me, guide me, if they are not showing any signs of any panic or whatever then it is fine as far as I am concerned.<br>P12-Rx-L, Int +6year, continued AM |

For key see table 2.
PCa, prostate cancer; PSA, prostate-specific antigen.

professionals, particularly family physicians, urologists and research nurses, about 'triggers' for active treatment. This inevitably highlighted the uncertainties surrounding the AM protocol (table 3 and figure 1). In parallel with the search for clarity and at times in direct response to the multiple uncertainties it revealed, men also sought control (figure 1). Ways of seeking control included choice of terminology to describe the cancer, influenced strongly by health professionals' language; who to tell about the diagnosis and when to discuss it. Superficially contradictory behaviours were motivated by this desire for control. During year 1 postdiagnosis, the AM protocol recommended 3-monthly testing. Men rapidly

**Figure 1** Strategies to manage uncertainties.

identified their 3-monthly PSA tests as a time when uncertainty and anxiety peaked and so developed contrasting ways to control their anxiety (table 3): some requested ongoing 3-monthly PSA tests, believing this gave better odds for controlling disease progression; others explicitly requested 6-monthly intervals or testing at the start of the month to control how often thoughts of PCa intruded. Some requested face-to-face appointments with the same health professional; others asserted control by plotting their own graphs of PSA values over time. Men (some with partner support) modified their exercise routine or diet or quit/reduced smoking/alcohol consumption. Crucially when men were supported to assert control confidence in the monitoring process grew and need for control dissipated (table 3, figure 1).

At the same time, this impetus to gain control was partly frustrated by ongoing uncertainties surrounding prognosis and the ability of the AM protocol to distinguish between indolent and progressing disease. Repeatedly facing this realisation, men described a parallel imperative to accept reassurances and invest trust in the healthcare team, particularly the urologist and research nurse. Men's accounts revealed a constant tension between strategies, oscillating between the search for clarity, control and reassurance. (table 3, figure 1). Although many men reported increased ability to accept reassurances with time, choice of strategy was also mediated by various other contextual factors.

## Contextual factors

A range of contextual factors were identified as mediating to what extent individuals accepted reassurances and developed trust in the healthcare team. The clearest contextual factor identified was time and all men reported that anxiety lessened with time on AM (table 4). Men who felt their needs for clarity and control were addressed, reported increased ability to place trust in the team with time. This was most evident in how men's preferences regarding timing and manner of regular PSA testing changed over time with support from health professionals: some came to accept longer intervals between tests or telephone consultations to replace face-to-face contact (table 3). Significant caring responsibilities gave men other priorities and made them less inclined to engage fully with monitoring or request radical treatments (table 4). Other contextual influences included the number of social or work commitments individual men were juggling, with busier men arguing they had little time to feel anxious about AM (table 4). In addition, close friends or family members with negative experiences of cancer could raise levels of concern or a sense that action was required to control the disease (table 4).

## Development of trust

Where men's needs for clarity, control and reassurance were met, trust in the healthcare team grew despite the inherent uncertainties and they continued with monitoring (tables 4 and 5). A trusting relationship with healthcare professionals was pivotal in maintaining this balance. Men who reported higher levels of anxiety at diagnosis and ongoing, but who nonetheless continued with AM throughout the study, all developed and retained a sense of trust in the healthcare team.

By contrast, failure to develop trust, or loss of that trust, usually resulting from unclear or contradictory messages, tended to lead men either to seek clarity by requesting a second opinion or, more commonly, reassert control by pressing for radical treatments or, in one case, withdrawing from monitoring (table 5). One man, who withdrew from

| Table 4 | Contextual factors |
|---|---|
| Influence of time in developing trust | Well, it's in your mind all the time you've got it like, you know, and was it going to flare up and as the years went on like and I'm still here, and I'm still here, I'm still here, you know, it got less like. That you know and well I'm now 78, I'm thinking, well, I've not done bad.<br>P1-Rx-L-8, Int +13yr5mo, radical treatment +13yr9mo |
| | Well not so bad really, you know I've had this condition now 6 years or so and you more or less get used to it, you get immune to it really. Yeah, so it's not so bad… It has got easier, you know when I first knew that I had it, it was a worry sort of thing you know, but it hasn't got so much a worry over the years.<br>P15-Rx-L-3, Int +8yr2mo, continued AM |
| | Over the three years it's been ok, and I've been, as far as I know they've really taken quite good care of me, they've given me plenty of time, I can ring up any time I want if I was worried about anything. I've got that, and I feel there's a contact there that I can ring and have a chat to somebody if I needed to.<br>P14-Rx-L,+2yr10mo, continued AM |
| Influence of caring role | She was wanting me to go and have this test done or something I don't know what it's called. I said to her not at the moment because I'm a bit busy and with the wife and everything, you know, it's been a bit awkward.<br>P18-Pref-L,+6yr2mo, withdrew from AM follow-up +7 year |
| Social context | I've got a pretty busy sort of life with my work and what I do within the village and stuff like that. So I ain't got time to brood on it, to be honest. So, yeah, get an active life might help some people. If you're not really, I can imagine if you were just sat around and not really a social sort of person or whatever and getting out and around and stuff, that you could sort of vegetate and brood on it a bit, perhaps.<br>P14-Pref-L, Int +4yr9mo, continued AM |
| | Well basically you don't think about it at all…well there's never a day that goes by you've got something to do, you know, two children, our two children, one is thirty odd and twenty-well nearly thirty. But like today I've just been down, his boiler's gone out, and so it goes on, there's always plenty to do<br>P9-Pref-L, Int+9mo, radical treatment +3yr4mo |
| Family and friends' experiences | I'm in a [name of club], we're all old people, I mean about 6 people got it [PCa]. And they're all sort of saying that their readings started to shoot up, so I thought I would nip it in the bud then, get it out early. All of them had treatment. Some of them had had surgery first, then radiotherapy, and for some of them that didn't work either so they're on hormones again. So I thought well, get it now, so I don't have to go through all this lot.<br>P9-Pref-L, Int +4yr5mo, radical treatment +3yr4mo |
| | I just lost my sister-in-law this week to breast cancer and it is horrible…It was really horrible, and I thought, 'I don't wanna go through that', I don't wanna be like that<br>P15-Rx-L, Int +2yr10mo, continued AM |

For key see table 2.

| Table 5 | Illustrations of loss or lack of trust |
|---|---|
| Conflicting advice | I was getting very confused…people saying you should have it [radical treatment] another one saying you shouldn't have it and another one saying you should have it and then one, I saw this radiology, radiologist whatever and he said 'well what do you think?' and I said 'what do I know?' I said, 'I'm seeing you, I'm expecting you to tell me!' and after a while he said 'well yes, perhaps you should then'. You know and you think well should I or shouldn't I? Really, I suppose my own [family] doctor at the time said, 'Well I would have it'.' P5-Rx-Unknown, Int +5yr5mo, radical treatment +4yr4mo |
| Lack of trust | So I mean really speaking it [AM] hasn't done a lot of good at all has it? So, I don't really know if this blood thing, this PSA does the trick or not. P18-Pref-L, Int +2yr5mo, withdrew from AM+7y |

For key see table 2.

ProtecT AM and the interview study at the time of his third interview, reported loss of trust in healthcare professionals, loss of confidence in the monitoring process and a wish to focus on caring for a family member. Two men requested radical treatment when it was not clinically indicated under the AM protocol; both reported contradictory messages from healthcare professionals and a lack of trust in the advice they received. These responses were an assertion of control in the context of loss of trust.

## DISCUSSION

This study of men's long-term experiences in following the ProtecT AM protocol for localised PCa found that men sought clarity, control and reassurance in order to deal with the uncertainties they faced, and developed trust in healthcare professionals when these needs were recognised and met. At the time of diagnosis, many sought clarity about 'triggers' for management change. They also sought control in apparently contradictory ways: over who knew their diagnosis, PSA test timing or manner, and over lifestyle choices. Given ongoing uncertainties, they sought reassurance from the clinical team and, where trust developed, monitoring became acceptable and normalised. Lack or loss of trust, arising, for example, through conflicting information or failure to accommodate attempts to control anxiety, led men to reassert control through, for example, requests for radical treatment or withdrawal from follow-up. Contextual factors influenced responses: caring responsibilities discouraged men from further investigations or radical treatment; friends or relatives with advanced cancer encouraged discussion about radical treatment. Crucially, where trust developed, even those experiencing higher levels of anxiety felt able to continue with AM where indicated.

This study is unique in reporting prospective data collected over a mean period of 7 (range 6–14) years, investigating experiences of a standardised AS/AM protocol, with a large dataset: 73 interviews among 20 men from 4 centres, each man interviewed at least three times. The longitudinal design captured how men's strategies and contextual factors changed with time and the pivotal role healthcare professionals could play in facilitating strategies and developing trust. The sample included men with low and intermediate/high-risk PCa, those randomly allocated to and those who chose AM, from a range of study sites, with a range of ages at diagnosis and both those continuing with AM and those undergoing radical treatments. More commonalities than differences were found between these groups.

Limitations include the fact that ProtecT study participants may have been positively oriented to AM, having accepted allocation to or chosen this pathway. They received RCT follow-up from ProtecT study research nurses and urologists involved in the study, rather than in routine clinics. It has been documented elsewhere that men valued the flexibility, accessibility and continuity of nurse-led AM[27]; this may have influenced the development of trust for the men in this study. Men outside the study, following different AS/AM protocols may report different experiences, although all face the same uncertainties regarding disease progression and need to balance control and trust. Only one man in this interview study reported withdrawing from AM; others withdrawing from AM may have had different experiences. Few men were from a non-white background.[25] All men started AM as their primary treatment. Of the nine patients who changed management, seven did so in response to clinical advice to initiate treatment in line with the AM protocol; only two patients changed management due to rising anxiety in the absence of such evidence. Most (n=15 or 75%) were diagnosed with low-risk PCa at baseline and around half (n=11) remained on AM throughout. Men with intermediate-risk/high-risk disease expressed similar views to those with low-risk disease. The overall sample size (20) limited scope for comparing experiences according to key criteria used to obtain a maximally diverse sample. However, comparison of experience across groups showed more commonalities than differences, meaning that findings reported here may be relevant to those following AM and potentially AS. In a parallel study,[30] men sought clarity about whether their radiotherapy treatment had been successful, but issues of control and trust had less salience in the longer term than they did for men on AM, who faced an ongoing need for clarity, control and trust to prevent a premature move off AM.

AS/AM has been recommended as the best option for men with low risk, clinically localised PCa in clinical

guidelines.[36] This is the first report to our knowledge, using longitudinal qualitative methods to illuminate men's experiences of undergoing this over such a long period. It highlighted men's need for clarity of information to support shared decision making about PCa treatment options and the benefits their wish for control may have in promoting engagement in shared decision making and healthy lifestyle choices as reported elsewhere.[37 38] Participants in this qualitative study reported raised anxiety, peaking at PSA testing, with varying intensity between men. ProtecT study patient-reported outcomes overall showed no greater anxiety among men randomised to AM than RP or RT at any time up to 6 years,[6] but these were summary measures of patient groups, and questionnaires were completed annually, not necessarily at the time of PSA testing. Previous qualitative research suggests that uncertainty and anxiety are issues for men on AM/AS[14 15 19] and may influence decisions to initiate radical treatments. Research has also suggested anxiety is highest at the time of diagnosis rather than during monitoring.[6 23 39] All men in this study acknowledged uncertainties and most acknowledged some anxiety at some point. However, it was notable that men reporting greater or persistent anxiety did not necessarily request change in management; anxiety could be managed successfully using coping strategies and reassurance from trusted health professionals. Previous research has shown men respond to surveillance by normalising, 'bracketing off' their cancer or striving to do 'something extra'[19 21] and has highlighted the importance of clarity of information and trust in the information provided for men.[37 38] These strategies and findings were replicated here, but a novel finding here was men's careful balancing of the need for clarity, control and reassurance and the key role of healthcare professionals in meeting these needs and developing trust.

Rapid increases in numbers of men receiving AS/AM for low-risk or low-volume intermediate risk PCa in the USA and UK[1 2] have occurred without evidence or consensus on inclusion criteria or optimal protocols.[18 28] Around 9% of patients change to radical treatment annually,[9 10 13] many without evidence of cancer progression.[7 11 13] This study indicates that men need care that respects their capacity to exercise control in line with their own coping strategies, and clear, consistent messages from health professionals are needed to develop and maintain trust. Health professionals supporting men following AS/AM should be aware of strategies that men adopt, the influence of contextual factors on men's decision making, and their own pivotal role in meeting men's needs and in building and retaining trust, if they are to tailor care to meet individual needs. Understanding the benefits to patients of clarity in communication and facilitating control where possible and an appreciation of the role of contextual factors will be essential for healthcare staff to build the trust required for long-term monitoring/surveillance.

Clear protocols for AM/AS are needed to provide men with information about indicators of disease progression and triggers for management change, as well as areas of uncertainty. In future, greater accuracy identifying indicators of disease progression may be provided by use of mpMRI and targeted biopsies to identify clinically significant disease. Further research is needed into the optimum ways of reassuring men in order to keep them engaged with AS/AM where appropriate.

## CONCLUSIONS

Men in the ProtecT RCT sought clarity, control and reassurance to accommodate anxiety over uncertainties of living with untreated PCa over a median of 7 years of AM. Where needs were met, men developed trust in healthcare professionals, this trust being essential in supporting men to continue with AM or change management when indicated. Clinicians can best build patient trust by being open and consistent in describing the certainties and uncertainties of the AM/AS protocol, allowing patients to exercise control where possible and being aware of contextual factors that may influence men's responses. Requests for radical treatment without evidence of cancer progression or withdrawal from AM were attempts to reassert control when trust failed: maintaining trust was pivotal in supporting men to stay with monitoring or change management when progression occurred. Robust follow-up protocols, including new developments in markers and imaging, to provide guidance for health professionals and patients about the detection of early disease progression, triggers for management change, and to provide confidence about when to continue with monitoring are under investigation. Insights from this study will remain critical for optimising clinician care that dovetails with individual men's needs and enables them to remain engaged with AS/AM.

**Author affiliations**
[1]Population Health Sciences, Bristol Medical School, University of Bristol, Bristol, UK
[2]Nuffield Department of Surgical Sciences, University of Oxford, Oxford, UK
[3]Department of Urology, University Hospitals of Leicester NHS Trust, Leicester, UK
[4]Department of Oncology and Metabolism, University of Sheffield Medical School, Sheffield, UK
[5]Department of Urology, Leeds Teaching Hospitals NHS Trust, Leeds, UK
[6]Department of Oncology and Metabolism, University of Sheffield, Sheffield, UK
[7]School of Medicine, Cardiff University, Cardiff, UK
[8]Department of Urology, North Bristol NHS Trust, Bristol, UK
[9]Department of Urology, Cardiff and Vale University Health Board, Cardiff, UK
[10]Department of Urology, NHS Lothian, Edinburgh, UK
[11]Faculty of Medicine, Health and Human Science, Macquarie University, Sydney, New South Wales, Australia
[12]Department of Urology, University Hospitals Birmingham NHS Foundation Trust, Birmingham, UK
[13]Department of Surgery, University of Cambridge, Cambridge, UK
[14]Department of Urology, Cambridge University Hospitals NHS Foundation Trust, Cambridge, UK
[15]Department of Urology, Newcastle Upon Tyne Hospitals NHS Trust, Newcastle Upon Tyne, UK

**Acknowledgements** Liz Salter and Dr Lucy Brindle contributed to data collection. We particularly wish to thank all men and their partners who participated in this

study, and other ProtecT study contributors: http://www.nejm.org/doi/suppl/10. 1056/NEJMoa1606221/suppl_file/nejmoa1606221_appendix.pdf

**Contributors** JW and JD designed the qualitative research. JW undertook the qualitative analysis and led the drafting of the paper. JD, AL, MD, EW, ET, RM, CM and TP contributed to the interpretation of the data. FH, JD and DN designed the ProtecT RCT and obtained the funding. ProtecT investigators RK, JC, AP, PH, DR, HK, ER, OH, PB, DG, AD, VJG and EP contributed to the acquisition of the data. All authors provided critical comments and approved the final version of the manuscript.

**Funding** This work was supported by the UK National Institute for Health Research (NIHR) Health Technology Assessment (HTA) Programme (projects 96/20/06 and 96/20/99, with the University of Oxford as sponsor (www.nets.nihr.ac.uk/projects/ hta/962099). JD, DN and FH are NIHR senior investigators. JD was previously a member of the NIHR Collaboration for Leadership in Applied Health Research and Care West, hosted by University Hospitals Bristol NHS Foundation Trust. FH is supported by the Oxford NIHR Biomedical Research Centre Surgical Innovation and Evaluation Theme and Cancer Research UK Oxford Centre. AL is supported by the Bristol Randomised Trials Collaboration, Bristol Trials Centre, University of Bristol, Bristol, UK. This study was designed and delivered in collaboration with the Bristol Randomised Trials Collaboration (BRTC), a UKCRC registered clinical trials unit which, as part of the Bristol Trials Centre, is in receipt of National Institute for Health Research CTU support funding.

**Disclaimer** The views and opinions expressed in this article are those of the authors and do not necessarily reflect those of the UK Department of Health and Social Care or the sponsor.

**Competing interests** None declared.

**Patient and public involvement** Patients and/or the public were involved in the design, or conduct, or reporting, or dissemination plans of this research. Refer to the Methods section for further details.

**Patient consent for publication** Not required.

**Ethics approval** Multicentre research ethics approval was granted by the UK East Midlands (formerly Trent) Multicentre Research Ethics Committee (01/4/025).

**Provenance and peer review** Not commissioned; externally peer reviewed.

**Data availability statement** Data are available on reasonable request. Data (anonymised interview transcripts) are available from JW at julia.wade@bristol.ac. uk on reasonable request and where costs of ensuring data are anonymised are met.

**ORCID iD**
Julia Wade http://orcid.org/0000-0001-6486-6477

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
