## [Reviewer comments · BMJ Open]

ARTICLE DETAILS

TITLE (PROVISIONAL)	Strategies adopted by men to deal with uncertainty and anxiety when following an active surveillance/monitoring protocol for localised prostate cancer and implications for care: a longitudinal qualitative study embedded within the ProtecT trial.
AUTHORS	Wade, Julia; Donovan, Jenny; Lane, Athene; Davis, Michael; Walsh, Eleanor; Neal, David; Turner, Emma; Martin, Richard; Metcalfe, Chris; Peters, Tim; Hamdy, Freddie; Kockelbergh, Roger; Catto, James; Paul, Alan; Holding, Peter; Rosario, Derek; Kynaston, Howard; Rowe, Edward; Hughes, Owen; Bollina, Prasad; Gillatt, David; Doherty, Alan; Gnanapragasam, Vincent; Paez, Edgar

VERSION 1 - REVIEW

REVIEWER	Andreas Dinkel Department of Psychosomatic Medicine and Psychotherapy, Klinikum rechts der Isar, School of Medicine, Technical University of Munich, Munich, Germany
REVIEW RETURNED	22-Dec-2019

GENERAL COMMENTS	This study investigates how men on active surveillance for prostate cancer adopt to feelings of uncertainty and anxiety. The authors conducted a qualitative study to identify strategies for managing uncertainty and anxiety. This study is unique as it uses a longitudinal design. All men were interviewed at least three time. 73 interviews were conducted over a median of 7 years. The results showed that men sought clarity and control to deal with uncertainties. Trust in the clinical team was essential for continuing surveillance. Furthermore, contextual factors, like responses from family and friends, influenced patient's decisions to continue surveillance. This is an interesting study providing further insight in the subjective experience of men undergoing active surveillance and factors impacting continuation/discontinuation of surveillance as an active treatment strategy. The longitudinal design makes this study unique. This allowed the researchers to uncover the pivotal role of trust in the clinical team for the long-term continuation of active surveillance and the detrimental effects of loss of trust. The introduction gives a concise overview of the topic. The methods are clearly stated. The results are presented in a clear fashion, the rich descriptions provided in the tables have been condensed nicely in the main text. In the discussion section, the authors provide a rich account of the relevance and the limitations of the results.
---

	I have only a few minor critical comments. 1. Methods, p. 8: The authors state that they used purposive sampling (p. 8). However, the guiding strategy is not stated explicitly. It seems that the authors' aim was to ensure maximum variation. Please comment. 2. Methods, p. 9: Data analysis was supported by qualitative data software. Please clarify - which steps of the analysis were supported by data software. 3. General comment: It seems reasonable that seeking clarity and control also would represent strategies of men with prostate cancer who chose prostatectomy or radiotherapy. Similarly, trust in the clinical team should be equally important for them as compared to men who chose active surveillance. - Maybe it would have been useful to include men who underwent prostatectomy or radiotherapy in the longitudinal qualitative analysis. The study design is unique, but are the results unique to men undergoing active surveillance?
--	--

REVIEWER	Kevin R. Loughlin Harvard Medical School Boston, Massachusetts U.S.A.
REVIEW RETURNED	07-Jan-2020

GENERAL COMMENTS	No meaningful conclusions can be drawn when only 20 patients are interviewed. The paper does not justify a length of 119 pages. There were only 4 centers involved, yet there are 24 authors. The senior authors need to determine if all the authors truly made a meaningful contribution to the manuscript. The study subjects do not represent a diverse population. There is the potential for selection bias throughout the manuscript.
--

REVIEWER	B. Heesterman IKNL, the Netherlands
REVIEW RETURNED	28-Jan-2020

GENERAL COMMENTS	This longitudinal serial in-depth qualitative interview study identifies coping strategies of men on active monitoring (AM) for localized prostate cancer (within the Protect trial). The authors describe that participants experience uncertainty about future disease progression and the ability of AM to detect progression. Coping strategies include seeking for clarity (about triggers for active treatment), control and accept reassurance. In addition, the authors describe that trust in healthcare professionals, mediated by contextual factors and the fulfillment of men's needs for clarity, control and reassurance, is essential to continue with AM. Comments [ ] Data was collected from 2000 until 2014. Why is there an approximately 5 year gap between data collection and submission? In the last two decades many studies have been published on the
--

	subject. Long-term outcomes of active surveillance have become available, predictive criteria for unfavorable pathologic outcome have been established and surveillance protocols have been updated accordingly. As stated in the discussion section 'men outside the study may report different experiences', considering the evolution of active surveillance, this may be particularly the case for patients currently on active surveillance. What is the relevance of this study for the current clinical practice? [ ] Are the participants included in this study representative for the population included in the Protect trial (e.g. what are the results of validated questionnaires and PROMS)? [ ] Considering that uncertainty regarding the ability of AM to detect disease progression is a concern for most participants, more elaborate description of the AM-protocol (in the body of the text) is justified. (page 7) [ ] What was the rationale for inviting only 25 men (in addition to the 5 men consenting at the time of PSA-testing)? Bearing in mind that the final number of 20 included participant is listed as a limitation of the study. (page 4 and 7) [ ] What was the rationale for including one patient with high-risk localized prostate cancer? Considering that active surveillance is currently not recommended for high-risk localized disease and patients with high-risk localized prostate cancer are generally not included in studies addressing active surveillance. (page 7 and 24) [ ] In the methods section it is stated that data between subgroups was compared. Although there is limited information regarding this analysis in the discussion section, the outcome of this analysis is not/scarcely reported in the results section. I suggest adding these results. (page 8, 15) [ ] Men with intermediate disease tended to change management sooner; data is not shown in table 1. I suggest adding these results. (page 10)
--	--

REVIEWER	Julia Menichetti University of Oslo, Norway
REVIEW RETURNED	05-Feb-2020

GENERAL COMMENTS	I endorse accepting this article and making it proceed in the revision process. I have some suggestions which can hopefully improve its quality. I think these are addressable if the authors are willing to work a little bit more on the analysis (or justify the analysis done). See file attached. BMJ OPEN 2019-036024–revision Thank you for asking me to review this interesting article about how prostate cancer patients in AS experience the AS strategy and deal with uncertainty over time. The study is particularly interesting given the longitudinal nature of the data collection. Findings provide
--

important insights about what can be helpful for these patients to manage the intrinsic uncertainty of AS.

I have some detailed suggestions which I think and hope can improve the quality of the manuscript.

1) Introduction

It would be valuable to provide more references and rationale for undergoing a longitudinal study. As it is framed now the introduction, the added value of having interviewed patients over time is not evident (even if there is, of course, as these patients may have to deal with anxiety and uncertainty over time, as they have biopsies every tot years, as they become older and become closer to death, etc...).

I also think the study aim can be improved, by focusing it more on “experiences of AM over time and strategies to manage uncertainty”, or maybe even “AM patients’ experiences of (how to deal with) uncertainty over time”

2) Methods

Some requests for more information:

a) I would like to have more info about the specificities of the AM strategy compared to e.g. AS or other monitoring strategies. These seem particularly relevant to understand the “uncertainties” these patients have to deal with and they received as information at the beginning. Also, I can see that also intermediate/high risk PCa patients were included in this arm.

b) Why only 4 over 9 centers were included in this qualitative study?

c) Was there a time limit in the inclusion of participants? Why “too much time passing since treatment began” was chosen as reason for exclusion?

d) Having done interviews both in person or by phone is a limitation, the modality changes so much this type of research and findings. How many were done by phone? Please discuss this in the discussion.

e) Does the interview track changed over time? Meaning, patients in different periods underwent different interviews and interviews covered different contents – those relevant for the patients at the moment of the interview? I imagine that this is the case, but it’s not clear...

f) The interview track is quite broad, including a lot of big topics – decision-making, experiences, views of treatment. Can you say something about this?

g) How were the participants purposively chosen? Based on what? What were the a priori choices in the sampling? I can see that the criterion was of maximum variability of the sample, but it is not clear how the authors applied it in the participants sampling and recruitment

h) It would be nice to have some more information about the analysis and about how researchers derived categories, themes and over themes from interviews (see also next comment)

3) Results

I am not completely convinced by the final themes and by how authors organized findings. I do think results are interesting, but I somehow find that the themes do not value the potential of insight that this study can bring. I wonder if a different organization of themes and findings would provide more useful insights to researchers and clinicians. As it is, it seems the focus is mostly about “how pt manage uncertainty during time”, with the loss/development of trust as result of a successful/unsuccessful process of managing the uncertainty of the AM and disease. Within this frame, I would see the contextual factors e.g. the experiences of friends/family as a possible contributor to how to manage uncertainty. It seems there are internal (the “strategies”), external (some of “the contextual factors”) and relational factors contributing to how pt learn to deal with uncertainty and live well the AM strategy. Or, differently, I wonder if the 3 strategies are actually the 3 key themes. In few words, I think themes and results can be organized in a clearer way. This is probably also related of seeing some themes, like the one of strategies, so rich and articulated, and others like the one of contextual factors quite poor. If this is not convincing, I would like at least to have some more information about how authors arrived to the final themes.

The subtheme “Influence of time in developing trust” it’s not convincing, being the aspect of time so intrinsic in the study and in the longitudinal data collection. I would say that authors should find a way to make emerge the development of discovered themes over time, and that time should be the leit motiv in the findings. Indeed, as it is now, one doesn’t get the feeling of how the experiences changed over time. I wonder if the aspect of time can be more explicitly mentioned throughout the findings, e.g. how do the strategies developed over time? Does some strategy play a greater role in the first AM phases etc...?

4) Discussion

i) I imagine that the AM experience of patients may have been influenced also by having done interviews over time (which may participants feel cared, receiving attention and interest by the study team), can you say something about that in the discussion?

j) I would include in the discussion the strategy/need to gain control, for its potentiality in supporting patients promoting their health and becoming more engaged in the care process and in healthy lifestyle changes (see Menichetti et al. “Making Active Surveillance a path towards health promotion: A qualitative study on prostate cancer patients’ perceptions of health promotion during Active Surveillance”). And the relationship between this strategy and trust.

k) I would also discuss more the clarity issue and connect it with the information needs of these patients (see Loeb et al. Informational needs during active surveillance for prostate cancer: A qualitative study)

REVIEWER	Megan McIntosh The University of Adelaide, Australia
REVIEW RETURNED	18-Feb-2020

GENERAL COMMENTS	Dear authors, Thank you for the opportunity to review your work. I think the longitudinal and qualitative approach are particularly interesting and important for this growing research area; more information on men's experiences on active surveillance and how they can be better supported, given its growing use in recent years, is undoubtedly required and this piece adds to this space nicely. In light of this, I have made some suggestions which I feel may strengthen the impact of this piece. I hope you find them useful and I look forward to seeing more research from the ProtecT trial in the future. 1. Quotes in tables Vs in the body I would strongly recommended including some, if not all, of the quotes within the body of the piece, rather than in the tables. In my opinion, it demonstrates the validity of the results and conclusions drawn more clearly, and the piece is therefore far easier to read and interpret. If the decision to put the quotes in tables was made to adhere to a word count, I would suggest negotiating with the editor to extend the wordcount to allow for this suggestion to be implemented. 2. Longitudinal results I would have liked to see more clear statements and discussions on the longitudinal results, given this format has not been used previously and is the main strength of the study. I acknowledge that some discussion of coping strategies changing over time was included in the results/discussion sections, however some additional evidence and discussion of this finding in particular would add greatly to the piece. Furthermore, in the introduction section you state that "longitudinal research is scarce", whilst I agree this is true, I would suggest adding a statement on the emerging findings of such research. For instance, https://doi.org/10.1002/pon.3657 and https://doi.org/10.1002/pon.5161 found anxiety decreases over time. 3. Active Surveillance Vs Active Monitoring Is there a reason these terms are used interchangeable throughout the article? I would suggest using the term Active Surveillance throughout, rather than changing between AS/AM or using AM, unless there are differences between the two. If so, please state these differences in the article. Using one term consistently will aid the efforts to minimise confusion and uncertainty regarding active surveillance versus other management options (e.g., watchful waiting). 4. Patients diagnosed with high-risk disease It is unclear why patients diagnosed with high-risk prostate cancer were included in the study, as active surveillance is not generally recommended for patients with intermediate risk – high risk disease. Please refer to the AUA/ASTRO/SUO 2017 Guidelines regarding active surveillance recommendations. It appears that at least 1 patient was diagnosed with high risk, and 3 with intermediate risk (+1 with an unknown risk level). Personally I feel that it is not appropriate to include the high-risk patient in the report, alternatively, a statement on why he was included and how/if his experiences may have been different or affected the results would be beneficial. 5. Withdrawal from monitoring Could you please clarify how many patients withdrew from monitoring in the duration of the study, and what withdrawal means? I assume this means they did not attend follow-up appointments for
---

	a period of time or indefinitely? Could this have affected the results or were any unique experiences identified that could be elaborated on? Do you have any suggestions on how health professionals might re-engage or better support men withdrawing from supervision? 6. Minor issues Below I have outlined some minor, mostly grammatical issues which I recommended attending to:  - Abstract: Capitalise 'Four' at the beginning of the settings heading. - Introduction:  o Paragraph 2, line 3: suggest saying "men can request radical treatment if they no longer tolerate this uncertainty and/or there is evidence of clinical progression". Men may transition to treatment for both reasons. o Paragraph 2, line 6: you don't use the acronym SRs anywhere else in the article, please remove. - Methods  o Please include a statement reflecting the average length of interviews - Discussion  o Paragraph 2, line 6: Please rewrite as the sample included low-high risk patients o Paragraph 2, line 9: Can you add a brief comment/summary on the commonalities & differences between the groups? o Paragraph 2, line 11: A comment on whether the nurse-led, urologist supported care as facilitated by the RCT may have affected experiences of the men interviewed would be beneficial. Would this differ from standard care? If so, the implications of this could be elaborated. o Paragraph 2, line 14: The sentence "all men started AM as their primary treatment, and all but two changed management only after evidence of progression" is confusing me, as n=11 participants were still on AM at the end of the study? Could you please clarify this sentence? o Paragraph 4: Suggest adding a statement reflecting the AUA guidelines (mentioned above) which recommend AS at the best treatment option for low risk PCa. o Paragraph 4: I would suggest adding a statement on the unmet needs men on AS (or men with PCA in general) experience. - Table 1:  o What do the two 'social class' categories reflect? Could these be better represented? o Are you able to provide any further information on the study centres (i.e., country)?
--	--

VERSION 1 – AUTHOR RESPONSE

Reviewer: 1

1. Methods, p. 8: The authors state that they used purposive sampling (p. 8). However, the guiding strategy is not stated explicitly. It seems that the authors' aim was to ensure maximum variation. Please comment.

This has been clarified on p8, lines 14-22 to read as follows:

Purposive sampling was used with the aim of obtaining a maximum variation sample for this interview study in terms of participant characteristics identified a priori as potentially impacting on experiences:

age, socio-economic background, whether randomly allocated to or choosing AM, and with low- or intermediate/high-risk disease. Recruitment took place at four of nine ProtecT centres to capture as diverse a range of experience as possible, whilst limiting time required for travel by the interviewer. Three centres were located across England and one was in Wales, UK.

2. Methods, p. 9: Data analysis was supported by qualitative data software. Please clarify - which steps of the analysis were supported by data software.

This has been clarified on p10, lines 19-21 to read as follows:

Data analysis was supported by qualitative data software NVivo.³² Coding carried out in NVivo enabled rapid retrieval of coded data for systematic comparison across participants and time.^{32,33}

3. General comment: It seems reasonable that seeking clarity and control also would represent strategies of men with prostate cancer who chose prostatectomy or radiotherapy. Similarly, trust in the clinical team should be equally important for them as compared to men who chose active surveillance. - Maybe it would have been useful to include men who underwent prostatectomy or radiotherapy in the longitudinal qualitative analysis. The study design is unique, but are the results unique to men undergoing active surveillance?

We thank the reviewer for highlighting this.

We have submitted a parallel paper, currently under review, that outlines longer term experiences of men following the radiotherapy pathway. As the reviewer has suggested, this paper found men were concerned about whether their radiotherapy treatment had been successful, but this was in the context of more general concerns about clear information about the expected impacts of radiotherapy over the short and longer term. Issues of balancing clarity, control and trust were considerably less salient in their reported experiences.

We have added the following sentence to address this issue in the discussion p19, lines 5-9:

In a parallel study³⁰ men sought clarity about whether their radiotherapy treatment had been successful, but issues of control and trust had less salience in the longer term than they did for men on AM, who face an ongoing need for clarity, control and trust to prevent a premature move off AM.

Reviewer: 2

1.No meaningful conclusions can be drawn when only 20 patients are interviewed.

This comment suggests unfamiliarity with qualitative research, and the comments are out of step with the other reviewers. The sample size is appropriate for a qualitative interview study and the sampling strategy follows guidelines for reporting qualitative research (O'Brien et al. 2014). The study uses a purposive sampling approach designed to produce a sample reflecting key factors that were identified as likely to impact on men's experiences. This has been clarified in the text on p8, lines 14-22 in response to Reviewer 1, comment 1 above.

2.The paper does not justify a length of 119 pages.

The text of the paper including Abstract, text, tables, figures and references numbers 38 pages, which is similar to other papers in the journal. Remaining pages are accounted for by supplemental files required by the journal.

3.There were only 4 centers involved, yet there are 24 authors. The senior authors need to determine if all the authors truly made a meaningful contribution to the manuscript.

All authors made an important contribution to the ProtecT study and this manuscript.

4. The study subjects do not represent a diverse population. There is the potential for selection bias throughout the manuscript.

Issues of representativeness and selection biases apply more to quantitative research designs. An outline of qualitative research methods, sampling, and qualitative generalisability can be found in Pope and Mays 1995 [Pope Catherine, Mays Nick. Qualitative Research: Reaching the parts other methods cannot reach: an introduction to qualitative methods in health and health services research BMJ 1995; 311 :42 <https://doi.org/10.1136/bmj.311.6996.42>]. This qualitative study follows the guidelines for reporting qualitative research (O'Brien et al. 2014, [https://doi: 10.1097/ACM.0000000000000388](https://doi.org/10.1097/ACM.0000000000000388)).

Reviewer: 3

1. Data was collected from 2000 until 2014. Why is there an approximately 5-year gap between data collection and submission?

Data analysis was held back until after the publication of the main 10-year analysis of the ProtecT study end of 2016. Funding limitations further constrained the completion of the manuscript.

2. In the last two decades many studies have been published on the subject. Long-term outcomes of active surveillance have become available, predictive criteria for unfavorable pathologic outcome have been established and surveillance protocols have been updated accordingly.

We thank the reviewer for these comments. While there have been an increasing number of studies, systematic reviews have shown that each study tends to have a different protocol for AS/AM, with different inclusion criteria, different monitoring strategies, and different triggers for change. There are also only small numbers of patients and cohorts with follow up longer than five years [Simpkin et al. 2015, Dall'Era et al. 2012, Dahabreh et al. 2012]. The ProtecT cohort is the largest with long-term follow up and a standardised protocol in a randomised trial. There have been very few cohort studies reporting patient reported outcome measures that have been able to distinguish AS/AM protocols from watchful waiting/observation or missing/no treatment [Chen et al. 2017, Barocas et al. 2017]. There have also been very few qualitative studies investigating experiences of monitoring or surveillance protocols, and most of these are cross-sectional or short-term [Kinsella et al. 2018, Bellardita et al. 2015]. We do not know of any published qualitative studies that have investigated men's experiences of monitoring over the length of time reported in this paper. This is stated p17, lines 17-20:

This study is unique in reporting prospective data collected over a mean period of 7 (range 6-14) years, investigating experiences of a standardised AS/AM protocol, with a large dataset: 73 interviews among 20 men from 4 centres, each man interviewed at least three times.

3. As stated in the discussion section 'men outside the study may report different experiences', considering the evolution of active surveillance, this may be particularly the case for patients currently on active surveillance. What is the relevance of this study for the current clinical practice?

We have outlined in comment 2 above how this study is relevant for current practice. In the paper, we wanted to acknowledge that ProtecT AM may have some differences from other AS protocols, particularly in relation to its management by trained research nurses in addition to senior specialists. However, this paper addresses a key issue that is part of any AM/AS protocol – that men need to balance the risk of disease progression against the wish to remain without radical treatment. This paper is the first to provide data from the perspective of the men involved, and with follow up over the short, medium and long term. We have amended the text on p18, lines 3-12 to reflect this:

Limitations include the fact that ProtecT study participants may have been positively oriented to AS/AM, having accepted allocation to or chosen this pathway. They received RCT follow-up from ProtecT study research nurses and urologists involved in the study, rather than through routine clinics. It has been documented elsewhere that men valued the flexibility, accessibility and continuity of nurse-led active monitoring;²⁵ this may have influenced the development of trust for the men in this study. Men outside the study, following different AS/AM protocols may report different experiences, although all face the same uncertainties regarding disease progression and need to balance control and trust.

4. Are the participants included in this study representative for the population included in the ProtecT trial (e.g. what are the results of validated questionnaires and PROMS)?

As indicated above, representativeness in a statistical sense is not the aim of qualitative research. However, the characteristics of men included in the interview study were broadly similar to the population included in the ProtecT trial and this has been made clear in the text on p 12, lines 7-9 as follows:

Comparison of baseline characteristics to those of men in the main ProtecT study³³ showed broad similarities: median age was 64yr (main study: 63yr); 9/20 (45%) were in managerial occupations (main study: 46%); 15/20 (75%) had low risk disease (main study: 72%).

The results of the PROMs for men undergoing AM have been published [Donovan et al. 2016]. We are completing an analysis of PROMs for treatments received, which includes analyses of men who actually received AM and who later changed management; this will be submitted in due course. The PROMs in ProtecT are wide-ranging and the analysis complex, so could not be included alongside other data in this paper.

5. Considering that uncertainty regarding the ability of AM to detect disease progression is a concern for most participants, more elaborate description of the AM-protocol (in the body of the text) is justified. (page 7)

The protocol for the ProtecT study has been published [Hamdy et al. 2016⁹ online supplement <https://www.nejm.org/doi/full/10.1056/NEJMoa1606220>]. A summary of the AM protocol has been added to the body of the text on p7, lines 7-13 as follows:

ProtecT AM aimed to allow men to avoid or delay radical treatment and its side-effects unless their prostate cancer progressed. It involved monitoring with PSA tests every 3 months for one year, then every 6 or 12 months thereafter, with an annual review. If PSA rose by 50% or more in a 12-month period or any concerns were raised by the patient or clinician at any time, there could be a re-evaluation of cancer status which could lead to continuing AM or changing to a radical option. This is described in detail elsewhere.^{25,9}

6. What was the rationale for inviting only 25 men (in addition to the 5 men consenting at the time of PSA-testing)? Bearing in mind that the final number of 20 included participant is listed as a limitation of the study. (page 4 and 7)

Thank you for raising this point. We have addressed it in the text as follows:

Longitudinal interview study (p7, line 24, p8, lines 2-6)

This study was part of a larger, longitudinal interview study, investigating men's experiences of treatment received within the ProtecT study and including participants following each of the treatment pathways (AM, RT, RP) with a median follow-up of 10 years.²⁹ In total 88 men were invited, and 65

men took part in one or more interviews. Experiences of men following a radical radiotherapy pathway have been submitted for publication elsewhere.³⁰

Participants (p8 lines 8-14)

Of those recruited to the longitudinal interview study, 20 men were recruited at diagnosis, chose or were randomly allocated to follow an AM pathway and took part in 3 or more interviews; their data are reported here. ProtecT study participants undergoing AM as their primary management following random allocation or treatment choice and within 12 months of PCa diagnosis, were eligible for this prospective interview study; this included men with clinically localised or intermediate/high-risk disease in line with eligibility criteria for the ProtecT RCT.²⁹

7. What was the rationale for including one patient with high-risk localized prostate cancer? Considering that active surveillance is currently not recommended for high-risk localized disease and patients with high-risk localized prostate cancer are generally not included in studies addressing active surveillance. (page 7 and 24).

We are aware that AS programs have very variable inclusion criteria which have changed over time [Simpkin 2015]. When this patient was identified as eligible, risk stratification was not used and he was included within the ProtecT study with clinically localised disease. As such he was eligible, invited and agreed to take part in the qualitative interview study. See also response to point 6 above.

8. In the methods section it is stated that data between subgroups was compared. Although there is limited information regarding this analysis in the discussion section, the outcome of this analysis is not/scarcely reported in the results section. I suggest adding these results. (page 8, 15)

We apologise that this wording may have led to a misunderstanding that this referred to quantitative subgroup analysis. This has been clarified in the Methods and Results as follows:

Methods (p10, lines 14-19)

Analysis proceeded by comparing data between groups within the sample to identify commonalities and contrasts of experiences. Groups included those accepting random allocation or choosing AM; remaining on AM or changing to radical treatment; with lower and higher risk disease; older or younger at diagnosis; and in earlier and later interviews to explore how views and experiences evolved over time.

Results (p12, lines 2-14)

Twenty participants (mean age at diagnosis 62, range 52-68 years) were interviewed a minimum three and maximum six times, with 73 interviews conducted over a median of 7 (range 6-14) years' follow-up. Nine (45%, 5 with low-risk disease) underwent radical treatment after a median of 4 years' (range 9mo-13yr9mo) follow-up, comparable with the ProtecT study⁹ Eleven (10 with low-risk disease) continued with AM for a median of 7 years' (range 6-7 years) follow-up (Table 1). Comparison of baseline characteristics to those of men in the main ProtecT study³⁵ showed broad similarities: median age was 64yr (main study: 63yr); 9/20 (45%) were in managerial occupations (main study: 46%); 15/20 (75%) had low risk disease (main study: 72%). Comparison of experiences across these groups showed more commonalities than differences, suggesting that the themes were relevant to men following AM.

9. Men with intermediate disease tended to change management sooner; data is not shown in table 1. I suggest adding these results. (page 10)

We thank the reviewer for this comment. These data will be reported in a separate paper looking at reasons why men on AM changed management and this sentence has therefore been edited to read as follows on p12, lines 14-18.

Five men with intermediate/high-risk disease expressed similar views to those with low-risk disease. There was a suggestion that higher risk men changed management sooner than those with low-risk disease, but this will be more comprehensively investigated in a future ProtecT study paper.

Reviewer: 4

I endorse accepting this article and making it proceed in the revision process. I have some suggestions which can hopefully improve its quality. I think these are addressable if the authors are willing to work a little bit more on the analysis (or justify the analysis done).

Thank you for asking me to review this interesting article about how prostate cancer patients in AS experience the AS strategy and deal with uncertainty over time. The study is particularly interesting given the longitudinal nature of the data collection. Findings provide important insights about what can be helpful for these patients to manage the intrinsic uncertainty of AS. I have some detailed suggestions which I think and hope can improve the quality of the manuscript.

We are very grateful for these positive comments.

1) Introduction

It would be valuable to provide more references and rationale for undergoing a longitudinal study. As it is framed now the introduction, the added value of having interviewed patients over time is not evident (even if there is, of course, as these patients may have to deal with anxiety and uncertainty over time, as they have biopsies every tot years, as they become older and become closer to death, etc...).

This is a very helpful suggestion and we have edited the Introduction on p5, lines 24 and p6 lines 1-6 as follows:

Although there is evidence that anxiety is minimal²² or decreases with time on AS,²³ longitudinal qualitative research on long-term experiences of AS remains scarce. Yet longitudinal studies offer an essential insight into how participants' needs and experiences may change over time, particularly as the disease develops slowly and patients may need to consider remaining on it or changing management over very many years. Longitudinal research also enables exploration of sensitive issues as the relationship between interviewer and interviewee develops.²⁴

I also think the study aim can be improved, by focusing it more on "experiences of AM over time and strategies to manage uncertainty", or maybe even "AM patients' experiences of (how to deal with) uncertainty over time"

Again, we appreciate this helpful suggestion. We have amended the Abstract and Introduction as follows: Abstract (p3, lines 4-7)

This study investigated men's strategies for coping with the uncertainties of active monitoring (AM, a surveillance strategy within the ProtecT trial) over the longer term and implications for optimising supportive care.

Background (p6, lines 8-17)

We addressed this evidence gap by conducting a longitudinal qualitative investigation of the experiences of men who followed the active monitoring (AM) protocol as their primary treatment in the

ProtecT (Prostate testing for cancer and Treatment) randomised controlled trial (RCT)²⁵ over a median of 7 (range 6-14) years. We investigated men's experiences of uncertainties and strategies to manage this. Findings provide insights to shape clinical services for the increasing numbers of patients now undergoing AS/AM programmes for clinically localised prostate cancer.

2) Methods

Some requests for more information:

a) I would like to have more info about the specificities of the AM strategy compared to e.g. AS or other monitoring strategies. These seem particularly relevant to understand the "uncertainties" these patients have to deal with and they received as information at the beginning. Also, I can see that also intermediate/high risk PCa patients were included in this arm.

We have provided more information as follows: response to Reviewer 3, point 5 above for more detail on the AM strategy; response to Reviewer 3, point 7 above for the rationale for including intermediate/high risk PCa patients.

We have also added the following sentence to the Methods p7, lines 18-20 (see response to Reviewer 3, point 2 above):

Uncertainties with AM were expected to be similar to those experienced with other AS strategies because of the lack of international consensus on protocols.²⁸

b) Why only 4 over 9 centers were included in this qualitative study?

Thank you for raising this point. We have now made this clearer in the text as follows (p8, lines 19-22):

Recruitment took place at four of nine ProtecT centres to capture as diverse a range of experience as possible, whilst limiting time required for travel by the interviewer. Three centres were located across England and one was in Wales, UK.

c) Was there a time limit in the inclusion of participants? Why "too much time passing since treatment began" was chosen as reason for exclusion?

First interviews were scheduled to take place approximately 6 months after men started the AM protocol and before a year had passed when they would have had at least 1 or 2 PSA tests. Men for whom 12 months on AM had already passed were no longer eligible to take part in this longitudinal qualitative study, which specifically aimed to capture experiences in the short, medium and long-term. This has been clarified in the text: (p8, lines 10-14 and p9 lines 4-7)

ProtecT study participants undergoing primary management AM, following random allocation or treatment choice and within 12 months of PCa diagnosis, were eligible for this prospective interview study; this included men with clinically localised or intermediate/high-risk disease in line with eligibility criteria for the ProtecT RCT.²⁹

Two men refused, four did not respond and six were ineligible due to receiving treatment outside the ProtecT study or because more than 12 months had passed since management by AM had begun.

d) Having done interviews both in person or by phone is a limitation, the modality changes so much this type of research and findings. How many were done by phone? Please discuss this in the discussion.

As the study was longitudinal, we wanted to be clear about how the interviews were done. We have now clarified how and why interviews were conducted in the Methods as follows: (p9, lines 14-18)
All initial interviews took place face-to-face to put interviewees at ease, in the location of their choice in homes, hospital outpatient departments or university premises as preferred. Subsequent

interviews were face-to-face or by telephone according to interviewee preference. Interviews lasted an average of 50 minutes (range 8-123 minutes).

e) Does the interview track changed over time? Meaning, patients in different periods underwent different interviews and interviews covered different contents – those relevant for the patients at the moment of the interview? I imagine that this is the case, but it's not clear...

f) The interview track is quite broad, including a lot of big topics – decision-making, experiences, views of treatment. Can you say something about this?

Apologies that this was not clear. The text has been amended as follows: (p9 lines 22-24, p10, line 1)

Interview topics are shown in Appendix A. These were derived from the literature prior to data collection but subsequently informed by data collection. Topic guides were tailored to fit experiences, for example, questions regarding experiences of radical treatments were included for those changing management.

g) How were the participants purposively chosen? Based on what? What were the a priori choices in the sampling? I can see that the criterion was of maximum variability of the sample, but it is not clear how the authors applied it in the participants sampling and recruitment

Please see response to Reviewer 1, point 1 above.

h) It would be nice to have some more information about the analysis and about how researchers derived categories, themes and over themes from interviews (see also next comment)

More information has been added to explain how analysis was conducted, please see page 10, lines 8-23 as follows: (See also responses to Reviewer 3, point 2 and Reviewer 3, point 8 above). Thematic analysis was used, applying principles of constant comparison based on grounded theory.³¹ Data collection and analysis proceeded iteratively, enabling preliminary findings to inform further data collection. Eight initial interview transcripts were analysed, coded and themes identified independently by JW and LS for subsequent cross-comparison of codes /themes, to refine the topic guide and inform further sampling and data collection. Disagreements were resolved by discussion to reach consensus. Analysis proceeded by comparing data between groups within the sample to identify commonalities and contrasts of experience according to group. Groups included those accepting random allocation or choosing AM; remaining on AM or changing to radical treatment; with lower and higher risk disease; older or younger at diagnosis; and comparing initial and later interviews to explore how views and experiences evolved over time. Data analysis was supported by qualitative data software NVivo.³² Coding carried out in NVivo enabled rapid retrieval of coded data for systematic comparison across participants and time.^{32,33} The standards for reporting qualitative research (SRQR) checklist³³ was applied in reporting this study. Interpretations of study data were regularly discussed by JW and JLD to counter risk of overinterpretation of findings.

3) Results

I am not completely convinced by the final themes and by how authors organized findings. I do think results are interesting, but I somehow find that the themes do not value the potential of insight that this study can bring. I wonder if a different organization of themes and findings would provide more useful insights to researchers and clinicians. As it is, it seems the focus is mostly about "how pt manage uncertainty during time", with the loss/development of trust as result of a successful/unsuccessful process of managing the uncertainty of the AM and disease. Within this

frame, I would see the contextual factors e.g. the experiences of friends/family as a possible contributor to how to manage uncertainty. It seems there are internal (the “strategies”), external (some of “the contextual factors”) and relational factors contributing to how pt learn to deal with uncertainty and live well the AM strategy. Or, differently, I wonder if the 3 strategies are actually the 3 key themes. In few words, I think themes and results can be organized in a clearer way. This is probably also related of seeing some themes, like the one of strategies, so rich and articulated, and others like the one of contextual factors quite poor. If this is not convincing, I would like at least to have some more information about how authors arrived to the final themes.

The subtheme “Influence of time in developing trust” it’s not convincing, being the aspect of time so intrinsic in the study and in the longitudinal data collection. I would say that authors should find a way to make emerge the development of discovered themes over time, and that time should be the leit motiv in the findings. Indeed, as it is now, one doesn’t get the feeling of how the experiences changed over time. I wonder if the aspect of time can be more explicitly mentioned throughout the findings, e.g. how do the strategies developed over time? Does some strategy play a greater role in the first AM phases etc...?

We appreciate this careful and detailed critique of how we have presented the results. We reflected on these comments and made edits on p12-16 to make clearer how we presented changes in experiences over this extended time period and have also drawn attention to Figure 1 to illustrate this (see examples given below). We have retained the current structure of presentation because this most closely reflects the way in which the themes and patterns emerged through the complex analysis comparing experiences between participants and over time, and where some changed management and others did not. We have also looked closely at our reporting of the contextual factors and made some further edits there as suggested (p14-15, examples below).

P13 lines 11-13:

Seeking clarity and control were strategies often employed early in the monitoring process. Men expressed a greater sense of reassurance as time passed, unless they lost confidence in the monitoring process (see below).

P14 lines 19-21

Although many men reported increased ability to accept reassurances with time, choice of strategy was also mediated by various other contextual factors.

P15, lines 1-4

A range of contextual factors were identified as mediating to what extent individuals accepted reassurances and developed trust in the healthcare team. The clearest contextual factor identified was time and all men reported that anxiety lessened with time on AM (Table 4).

P15 lines 5-9

This was most evident in how men’s preferences regarding timing and manner of regular PSA testing changed over time with support from health professionals: some came to accept longer intervals between tests or telephone consultations to replace face-to-face contact (Table 3).

P15 lines 10-15

Other contextual influences included the number of social or work commitments individual men were juggling, with busier men arguing they had little time to feel anxious about AM (Table 4). In addition, close friends’ or family members with negative experiences of cancer could raise levels of concern or a sense that action was required to control the disease Table 4).

P16 lines 7-11

Two men requested radical treatment when it was not clinically indicated under the AM protocol; both reported contradictory messages from healthcare professionals and a lack of trust in the advice they received. These responses were an assertion of control in the context of loss of trust.

4) Discussion

i) I imagine that the AM experience of patients may have been influenced also by having done interviews over time (which may participants feel cared, receiving attention and interest by the study team), can you say something about that in the discussion?

Thank you - we have added a comment about this point to our discussion: (p16)

The process of taking part in the interview study itself may have increased men's sense of being cared for and influenced responses reported here; however, not all men reported positive experiences.

j) I would include in the discussion the strategy/need to gain control, for its potentiality in supporting patients promoting their health and becoming more engaged in the care process and in healthy lifestyle changes (see Menichetti et al. "Making Active Surveillance a path towards health promotion: A qualitative study on prostate cancer patients' perceptions of health promotion during Active Surveillance"). And the relationship between this strategy and trust.

k) I would also discuss more the clarity issue and connect it with the information needs of these patients (see Loeb et al. Informational needs during active surveillance for prostate cancer: A qualitative study).

Again, we found these very helpful suggestions and we have referenced these studies in our discussion as follows:

P19 lines 11-17

AS/AM has been recommended as the best option for men with low risk clinically localised PCa in clinical guidelines.³⁶ This is the first report to our knowledge, using longitudinal qualitative methods to illuminate men's experiences of undergoing this over such a long period. It highlighted men's need for clarity of information to support shared decision making about PCa treatment options and the benefits their wish for control may have in promoting engagement in shared decision making and healthy lifestyle choices as reported elsewhere.^{37,38}

P20, lines 8-13

Previous research has shown men respond to surveillance by normalizing, 'bracketing off' their cancer or striving to do 'something extra'^{19,21} and has highlighted the importance of clarity of information and trust in the information provided for men.^{37,38} These strategies and findings were replicated here, but a novel finding here was men's careful balancing of the need for clarity, control and reassurance and the key role of healthcare professionals in meeting these needs and developing trust.

Reviewer: 5

Thank you for the opportunity to review your work. I think the longitudinal and qualitative approach are particularly interesting and important for this growing research area; more information on men's experiences on active surveillance and how they can be better supported, given its growing use in recent years, is undoubtedly required and this piece adds to this space nicely. In light of this, I have made some suggestions which I feel may strengthen the impact of this piece. I hope you find them useful and I look forward to seeing more research from the ProtecT trial in the future.

Thank you very much for these very positive comments.

1. Quotes in tables Vs in the body

I would strongly recommended including some, if not all, of the quotes within the body of the piece, rather than in the tables. In my opinion, it demonstrates the validity of the results and conclusions drawn more clearly, and the piece is therefore far easier to read and interpret. If the decision to put the quotes in tables was made to adhere to a word count, I would suggest negotiating with the editor to extend the wordcount to allow for this suggestion to be implemented.

Thank you for this suggestion. Normally our preference would be to present quotes as suggested here. Our decision to present the quotes in tables was because it better suits the journal style.

2. Longitudinal results

I would have liked to see more clear statements and discussions on the longitudinal results, given this format has not been used previously and is the main strength of the study. I acknowledge that some discussion of coping strategies changing over time was included in the results/discussion sections, however some additional evidence and discussion of this finding in particular would add greatly to the piece. Furthermore, in the introduction section you state that "longitudinal research is scarce", whilst I agree this is true, I would suggest adding a statement on the emerging findings of such research. For instance, <https://doi.org/10.1002/pon.3657> and <https://doi.org/10.1002/pon.5161> found anxiety decreases over time.

Thank you, this is a particularly helpful point. We now draw attention to Figure 1 which illustrates how strategy-use changed over time and we have amended the text of the Results to highlight how coping strategies changed with time. Please also see our response to Reviewer 4, point 3 above.

We have also amended the Introduction to include these 2 references:

P5 line 24 and P6 line 1-2

Although there is evidence that anxiety is minimal²² or decreases with time on AS,²³ longitudinal qualitative research on long-term experiences of AS remains scarce.

3. Active Surveillance Vs Active Monitoring

Is there a reason these terms are used interchangeable throughout the article? I would suggest using the term Active Surveillance throughout, rather than changing between AS/AM or using AM, unless there are differences between the two. If so, please state these differences in the article. Using one term consistently will aid the efforts to minimise confusion and uncertainty regarding active surveillance versus other management options (e.g., watchful waiting).

We hope this is now clearer. We have clarified use of the term active monitoring (AM) when first introduced on p6 as the name of the active surveillance strategy used within the ProtecT study. Elsewhere we refer to AS/AM to refer to the range of active surveillance protocols used more generally.

4. Patients diagnosed with high-risk disease

It is unclear why patients diagnosed with high-risk prostate cancer were included in the study, as active surveillance is not generally recommended for patients with intermediate risk – high risk disease. Please refer to the AUA/ASTRO/SUO 2017 Guidelines regarding active surveillance recommendations. It appears that at least 1 patient was diagnosed with high risk,

and 3 with intermediate risk (+1 with an unknown risk level). Personally I feel that it is not appropriate to include the high-risk patient in the report, alternatively, a statement on why he was included and how/if his experiences may have been different or affected the results would be beneficial.

Please see the response to Reviewer 3, points 6 and 7 above.

5. Withdrawal from monitoring

Could you please clarify how many patients withdrew from monitoring in the duration of the study, and what withdrawal means? I assume this means they did not attend follow-up appointments for a period of time or indefinitely? Could this have affected the results or were any unique experiences identified that could be elaborated on? Do you have any suggestions on how health professionals might re-engage or better support men withdrawing from supervision?

Thank you for raising this point as we should clarify this. Only one man included in this interview study reported in his third and final interview that he was not continuing with ProtecT follow up or the interview study for a combination of reasons. Given that he was the only man, we do not feel we can draw conclusions about this:

P16, lines 4-7:

One man, who withdrew from ProtecT AM and the interview study at the time of his third interview, reported loss of trust in healthcare professionals, loss of confidence in the monitoring process and wanted to focus on caring for a family-member.

6. Minor issues

Below I have outlined some minor, mostly grammatical issues which I recommended attending to:

- Abstract: Capitalise 'Four' at the beginning of the settings heading.
- Introduction:
 - o Paragraph 2, line 3: suggest saying "men can request radical treatment if they no longer tolerate this uncertainty and/or there is evidence of clinical progression". Men may transition to treatment for both reasons.
 - o Paragraph 2, line 6: you don't use the acronym SRs anywhere else in the article, please remove.

All changed as suggested. The acronym SR was retained as it now used later on the same page.

- Methods

- o Please include a statement reflecting the average length of interviews

We have given the average length and range of length of interviews:

P9, lines 18

Interviews lasted an average of 50 minutes (range 8-123 minutes).

- Discussion

- o Paragraph 2, line 6: Please rewrite as the sample included low-high risk patients
- o Paragraph 2, line 9: Can you add a brief comment/summary on the commonalities & differences between the groups?

This has been changed as suggested:

P17, lines 22-24 and p18, lines 1-2:

The sample included men with low and intermediate/high risk PCa, those randomly allocated to and those who chose AS/AM, from a range of study sites, with a range of ages at diagnosis, and those continuing with AS/AM or undergoing radical treatments. More commonalities than differences were found between these groups.

o Paragraph 2, line 11: A comment on whether the nurse-led, urologist supported care as facilitated by the RCT may have affected experiences of the men interviewed would be beneficial. Would this differ from standard care? If so, the implications of this could be elaborated.

Please see our response to Reviewer 3, point 3 and text on p18, lines 4-10 in response:

They received RCT follow-up from ProtecT study research nurses and urologists involved in the study, rather than in routine clinics. It has been documented elsewhere that men valued the flexibility, accessibility and continuity of nurse-led AM;²⁷ this may have influenced the development of trust for the men in this study. Men outside the study, following different AS/AM protocols may report different experiences, although all face the same uncertainties regarding disease progression and need to balance control and trust.

o Paragraph 2, line 14: The sentence “all men started AM as their primary treatment, and all but two changed management only after evidence of progression” is confusing me, as n=11 participants were still on AM at the end of the study? Could you please clarify this sentence?

This section has now been edited to read as follows:

P18, lines 13-19

All men started AM as their primary treatment. Of the nine patients who changed management, seven did so in response to clinical advice to initiate treatment in line with the AM protocol; only two patients changed management due to rising anxiety in the absence of such evidence. Most (n=15 or 75%) were diagnosed with low-risk PCa at baseline and around half (n=11) remained on AM throughout.

o Paragraph 4: Suggest adding a statement reflecting the AUA guidelines (mentioned above) which recommend AS at the best treatment option for low risk PCa.

A sentence has been added to this paragraph as follows:

P19, lines 11-12

AS/AM has been recommended as the best option for men with low risk, clinically localised PCa in clinical guidelines³⁶.

o Paragraph 4: I would suggest adding a statement on the unmet needs men on AS (or men with PCA in general) experience.
Please see response to Reviewer 4, point j above.

Table 1:

o What do the two ‘social class’ categories reflect? Could these be better represented?

The social class categories reflect a simple division into managerial/professional vs others, as has been used elsewhere in the study [Neal et al. 2019 <https://doi.org/10.1016/j.eururo.2019.10.030> and Donovan et al. 2019, <https://doi.org/10.1016/j.jclinepi.2019.05.036>]

o Are you able to provide any further information on the study centres (i.e., country)?

Please see response to Reviewer 4, 2b above and text inserted on p8, lines 21-22.

VERSION 2 – REVIEW

REVIEWER	Andreas Dinkel Department of Psychosomatic Medicine and Psychotherapy, Klinikum rechts der Isar, School of Medicine, Technical University of Munich, Germany
REVIEW RETURNED	26-Apr-2020

GENERAL COMMENTS	Thank you for your efforts in revising the manuscript. The authors have addressed all of my concerns.
---

REVIEWER	B. Heesterman IKNL, the Netherlands
REVIEW RETURNED	06-May-2020

GENERAL COMMENTS	The authors have sufficiently addressed my concerns.
--

REVIEWER	Julia Menichetti University of Oslo, Norway
REVIEW RETURNED	21-Apr-2020

GENERAL COMMENTS	I think the authors have made a good (successful) effort in addressing all the reviewers comments, and I think the paper has significantly improved. I still think that findings could have been organised more clearly and lifted a little bit more up, but I understand the authors point of being as much as possible adherent to the data and what emerges from the data.
---

REVIEWER	Megan McIntosh University of Adelaide, Australia
REVIEW RETURNED	24-Apr-2020

GENERAL COMMENTS	I feel the authors have done an excellent job in refining the manuscript based on the 5 reviewer comments, including my own. I believe this paper is now in an appropriate form for publication and I have no further comments. I strongly support the publication of this study, and look forward to seeing it available to other researchers and readers of BMJ Open.
---